# Relaxing Representation Alignment with Knowledge Preservation for Multi-Modal Continual Learning

## Abstract

In continual learning, developing robust representations that adapt to new distributions or classes while retaining prior knowledge is crucial. While most traditional approaches focus on single-modality data, multi-modal learning offers significant advantages by leveraging diverse sensory inputs, akin to human perception. However, transitioning to multi-modal continual learning introduces additional challenges as the model needs to effectively combine new information from different modalities while avoiding catastrophic forgetting. In this work, we propose a relaxed cross-modality representation alignment loss and utilize a dual-learner framework to preserve the relation between previously learned representations. We validate our framework using several multi-modal datasets that encompass various types of input modalities. Results show that we consistently outperform baseline continual learning methods in both class incremental and domain incremental learning scenarios. Further analysis highlights the effectiveness of our solution in preserving prior knowledge while incorporating new information.

## 1 Introduction

Developing robust representations that can adapt to new distributions or classes is crucial when continual learning on a sequence of classification tasks. At the heart of this is representation learning, which ensures that models not only adapt to new information but also retain previously learned knowledge effectively. While most traditional continual learning approaches focus on single-modality data (Chaudhry et al., 2019; Buzzega et al., 2020; Kang et al., 2022; Sarfraz et al., 2023; Shi & Wang, 2024), multi-modal data offers significant advantages. This mirrors how human sensory systems combine diverse inputs to enhance understanding and improve predictive accuracy.

However, extending continual learning to multi-modal settings introduces additional complexities. Existing frameworks, such as traditional contrastive learning (Radford et al., 2021; Jia et al., 2021), have shown success in multi-modal scenarios by aligning different modalities into a unified representational space. Adapting these frameworks to multi-modal continual learning presents two key challenges. First, strict cross-modality alignment can limit the model's ability to capture modality-specific features (Jiang et al., 2023), potentially missing important details such as auditory cues not present in visual data. As continual learning progresses, the reliance on a constrained feature set can lead to overlaps between representations from different classes and degrades performance on previously learned tasks. Figure 1 illustrates this issue using two instrument classes from AVE (Tian et al., 2018), Guitar and Ukelele, with high visual resemblance but have different tones. We observe an overall increase in similarity scores between representations of the two classes as learning progresses, indicating a greater overlap between the two classes. Second, current contrastive methods lack a clear mechanism to preserve earlier representations while enhancing multi-modal learning.

To address these challenges, we propose a novel approach that relaxes the cross-modality alignment constraint. Rather than enforcing direct alignment between modalities, we independently align each modality's representation with a joint representation formed by fusing the various modalities. Since the joint representation encapsulates information from all modalities, aligning each modality with it minimizes information loss, thereby results in more stable representations. Additionally, we apply a

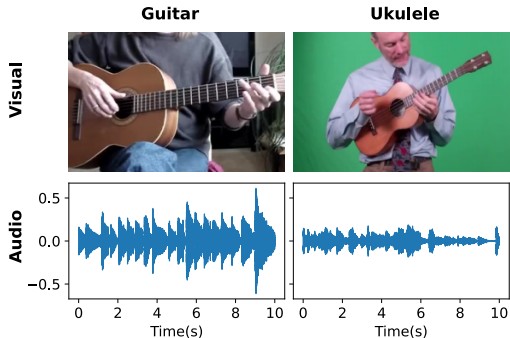

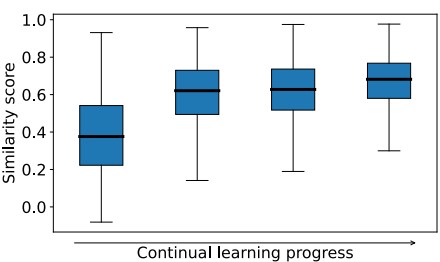

Figure 1: (Left) Sample video frame and audio of guitar and ukelele class from AVE. (Right) Distribution of similarity score between representations of the two classes as learning progresses.

regularization term at each incremental learning step that penalizes changes in the relation between previously learned representations to mitigate the issue of catastrophic forgetting.

We develop a general multi-modal continual learning framework applicable across diverse input modalities. To validate our approach, we conduct experiments on several multi-modal classification datasets under varied continual learning scenarios such as class incremental and domain incremental learning. Experimental results demonstrate significant performance improvements across these scenarios. Further analysis shows that our solution achieves good performance in retaining old knowledge while acquiring new information.

## 2 RELATED WORK

Traditional continual learning methods typically stores a small set of samples from previous tasks for replay and regulating model updates to mitigate forgetting. Experience Replay (ER) (Riemer et al., 2019) interleaves memory samples with the current task samples during training. GEM (Lopez-Paz & Ranzato, 2017) and A-GEM (Chaudhry et al., 2019) constrain the model update to be in the direction orthogonal to the gradients with respect to loss on memory samples. DER++ (Buzzega et al., 2020) uses an additional distillation loss to ensure the output logits of the memory samples remains consistent. Co$^2$L (Cha et al., 2021) uses contrastive learning to learn transferable representations and preserves the relation between representation of memory samples to reduce forgetting. SS-IL (Ahn et al., 2021) uses separate distillation loss on logits of classes learned at different tasks to reduce bias to new classes. AFC (Kang et al., 2022) uses importance-weighted distillation loss on features to minimize the upper bound of loss increase on previous tasks. ESMER (Sarfraz et al., 2023) updates the model using low-loss samples to reduce abrupt representation drift. UDIL (Shi & Wang, 2024) allows learnable coefficients to balance the loss terms on samples of current task and memory samples. All these methods focus on single image modality and do not take into account learning of the correlation between different modalities.

Meanwhile, research on multi-modal representation learning aims to learn robust representations capturing information from multiple input modalities. CLIP (Radford et al., 2021) and ALIGN (Jia et al., 2021) uses contrastive loss to align the image and text representations within a unified representational space. However, Jiang et al. (2023) show that exact alignment across modalities may be sub-optimal for downstream tasks, and propose to differentiate representations into components that capture modality-specific and modality-shared information separately. Similarly, SimMMDG (Dong et al., 2024) employs a representation splitting approach and utilizes label information to apply supervised contrastive loss only on the modality-shared component. In contrast to these methods, which assume data to be independent and identically distributed, our focus is to learn robust multi-modal representation under the continual learning setting, where data of different distributions or classes become available over time.

Some recent works focus on continual learning for multi-modal classification tasks to address the issue of integrating and retaining knowledge from different modalities over time. AV-CIL (Pian et al., 2023) focuses on audio-visual class incremental learning and preserves the semantic similarity

between the two modalities by maximizing similarity between cross-modal features of the same class while minimizing those of different classes. It utilizes distillation loss on the visual attention maps to preserve the model's attentive ability. CIGN (Mo et al., 2023) also focuses on audio-visual continual learning and uses learnable audio-visual class tokens in the transformer architecture to capture class-aware features. It introduces a distillation loss to preserve the distribution of previously learned class tokens. CMR-MFN (Wang et al., 2023) examines continual learning in egocentric activity recognition using visual-inertial data. This work employs a confusion mixup strategy and dynamic expandable architecture to adaptively manage the changing correlations between modalities over time. All these methods rely on specific architectures and modality configurations, which may limit their applicability across broader multi-modal continual learning settings.

## 3 METHODOLOGY

We first define the problem setup of continual learning involving a sequence of $T$ i.i.d. multi-modal classification tasks. At each incremental step $t \in [1, T]$, the learner is given a dataset $\mathcal{D}_t = \{(\boldsymbol{x}_i, y_i)\} \sim P_t$, where $P_t$ is the distribution of the $t$-th task. Each data instance $\boldsymbol{x}_i = \{\boldsymbol{x}_i^k\}_{k=1}^K$ comprises of $K$ different input modalities and $y_i \in \mathcal{Y}_t$ is its corresponding label with $\mathcal{Y}_t$ denoting the set of classes seen at the $t$-th incremental step. Note that except for a small number of those retained in memory $\mathcal{M}_t$, data from previous tasks are not accessible.

Training at each incremental step $t$ utilizes $\mathcal{D}_t' = \mathcal{D}_t \cup \mathcal{M}_t$, a combination of the current task's data and retained memory instances. After training, a maximum of $m$ data instances are sampled from $\mathcal{D}_t'$ to create $\mathcal{M}_{t+1}$, the memory for the next incremental step. Given the dataset $\mathcal{D}_t'$, the goal is to learn a model that minimizes prediction errors on test samples drawn from the joint distribution of all the tasks seen so far.

### 3.1 RELAXING REPRESENTATION ALIGNMENT CONSTRAIN

Multi-modal representation learning plays an important role in multi-modal continual learning. Existing works (Radford et al., 2021; Jia et al., 2021) have shown the effectiveness of using contrastive loss to project multi-modal representations into a common feature space for more robust representations. Supervised contrastive loss (Khosla et al., 2020) leverages the label information to cluster representations of the same class together while pushing apart clusters of different classes. Multi-modal supervised contrastive loss then aligns representations of instances from different modalities with the same labels. However, initial experiments employing the contrastive loss in multi-modal continual learning reveal a rapid degradation in model performance as learning progresses (see Appendix A1 for details). This is because different modalities inherently capture distinct information and the strict constraint to align representations from different modalities into a common feature space imposed by the contrastive loss often leads to the loss of modality-specific information (Dong et al., 2024), thereby accelerating the process of forgetting. This motivates us to design a new representation alignment loss that relaxes the constrain and encourages the model to retain the distinct features captured by the different modalities.

Suppose we have a batch of $N$ training samples from $\mathcal{D}_t'$ with $\mathcal{B}_{\mathcal{D}_t}, \mathcal{B}_{\mathcal{M}_t} \subset [1, N]$ being the set containing indices of samples from $\mathcal{D}_t$ and $\mathcal{M}_t$ respectively. For each sample $\boldsymbol{x}_i$, let $\boldsymbol{z}_i^k$ be the corresponding projected representation of input modality $\boldsymbol{x}_i^k$ and $\boldsymbol{z}_i$ be the joint representation obtained from combining all the modalities. We collect all joint representations in the batch to form the set $\mathcal{J} = \{\boldsymbol{z}_j \mid j \in [1, N]\}$. For each modality-specific representation $\boldsymbol{z}_i^k$ acting as an anchor, we define the set comprising all representations of the same modality $k$, except for the anchor itself, as $\mathcal{A}_i^k = \{\boldsymbol{z}_j^k \mid j \in [1, N], j \neq i\}$. We identify the 'positives' of the anchor as those representations in $\mathcal{J}$ and $\mathcal{A}_i^k$ that belong to samples with the same class labels as $\boldsymbol{x}_i$. The corresponding sets of positives for the joint representation and for each modality-specific representation are given by $\mathcal{J}_i = \{\boldsymbol{z}_j \in \mathcal{J} \mid y_i = y_j\}$ and $\mathcal{Q}_i^k = \{\boldsymbol{z}_j^k \in \mathcal{A}_i^k \mid y_i = y_j\}$ respectively. The remaining representations in $\mathcal{J} \setminus \mathcal{J}_i$ and $\mathcal{A}_i^k \setminus \mathcal{Q}_i^k$ are then 'negatives' of the anchor. The goal is to pull together the anchor and positives, while pushing apart the anchor from negatives.

To reduce loss of modality-specific information resulting from cross-modality alignment, we exclude pairs with different modalities when defining the positives and negatives of each modality-

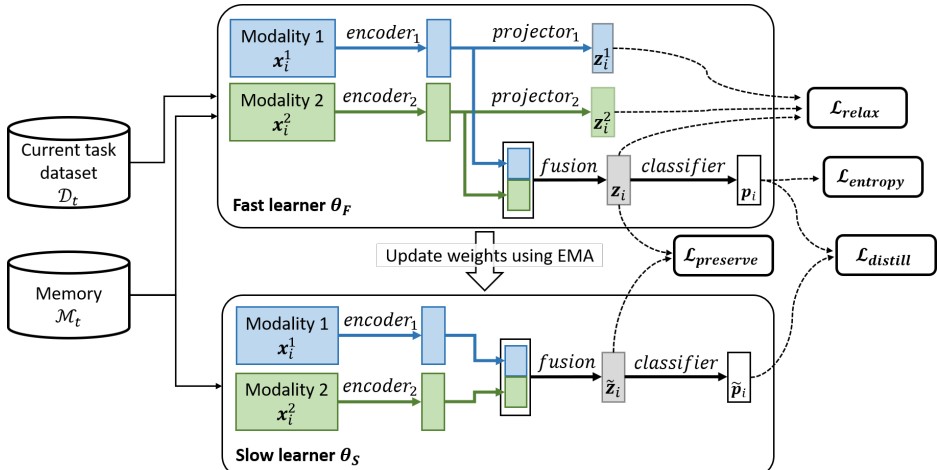

Figure 2: Overview of our framework.

specific representations $z_i^k$ that acts as anchor. Instead, we form pairs with the joint representations $z_i$. By independently aligning each modality-specific representation to the joint representations, we indirectly align all modalities into a unified space while ensuring minimal loss of modality-specific information since joint representations encapsulates combination of information from all modalities. The loss for relaxing the alignment of representations across different modalities is then defined as:

$$\mathcal{L}_{relax} = \frac{1}{K|\mathcal{B}_{\mathcal{D}_t}|} \sum_{i \in \mathcal{B}_{\mathcal{D}_t}} \sum_{k=1}^{K} \frac{-1}{|\mathcal{Q}_i^k|} \sum_{q \in \mathcal{Q}_i^k \cup \mathcal{J}_i} \log \frac{\exp(z_i^k \cdot q/\tau)}{\sum_{a \in \mathcal{A}_i^k \cup \mathcal{J}} \exp(z_i^k \cdot a/\tau)}, \quad (1)$$

where $\tau > 0$ is a scalar temperature hyperparameter and $\cdot$ denotes the cosine similarity between two normalized representations.

## 3.2 DUAL LEARNER FRAMEWORK

Relaxing the alignment constrain reduces the model's tendency to bias towards learning modality-shared features for classifying new tasks. Although $\mathcal{L}_{relax}$ promotes the learning of more stable representation, it does not explicitly mitigate the issue of forgetting previously learned tasks. To address this issue, we leverage the theory of complementary learning systems which posits that the brain uses two specialized systems to achieve effective learning (McClelland et al., 1995; Kumaran et al., 2016). Specifically, the hippocampus plays the role in fast learning of specifics of individual experiences, while the neocortex relies on slow learning to gradually form structured knowledge about the environment. Inspired by this neuropsychological framework, various models have been proposed that employ complementary learners to simulate these fast and slow learning dynamics for effective continual learning (Rostami et al., 2019; Pham et al., 2021; Arani et al., 2022; Sarfraz et al., 2023). This motivates us to adopt a dual-learner framework for multi-modal continual learning.

Figure 2 shows our dual-learner framework, consisting of a fast learner which quickly integrates new knowledge from current task, and a slow learner which gradually accumulates knowledge from the fast learner. Both learners have the same modality-specific encoders that independently process each input modality $x_i^k$. The output from the encoders are concatenated and forwarded to the fusion layers to obtain a joint representation $z_i$, which is then used by the classifier to output logits $p_i$. The fast learner also have a projection layer for each modality that projects output from the respective encoders to obtain representations $z_i^k$ with the same dimension as $z_i$.

We use both the fast and slow learners to extract the joint representation of each memory sample in the batch, denoted as $z_i$ and $\widetilde{z}_i$ respectively for $i \in \mathcal{B}_{\mathcal{M}_t}$. To quantify the relation between these samples, we compute the normalized pairwise similarity between the representations from each learner as follows:

$$s_{i,j} = \frac{\exp(z_i \cdot z_j/\tau)}{\sum_{l \in \mathcal{B}_{\mathcal{M}_t} \setminus \{i\}} \exp(z_i \cdot z_l/\tau)} \qquad \widetilde{s}_{i,j} = \frac{\exp(\widetilde{z}_i \cdot \widetilde{z}_j/\tau)}{\sum_{l \in \mathcal{B}_{\mathcal{M}_t} \setminus \{i\}} \exp(\widetilde{z}_i \cdot \widetilde{z}_l/\tau)}. \quad (2)$$

To ensure that the relation among memory samples remain consistent as new information is integrated, we regulate the model updates using the loss minimizing Kullback–Leibler divergence between the similarity scores obtained from the fast and slow learners as follows:

$$\mathcal{L}_{preserve} = \frac{\lambda}{|\mathcal{B}_{\mathcal{M}_t}|} \sum_{i \in \mathcal{B}_{\mathcal{M}_t}} \sum_{j \in \mathcal{B}_{\mathcal{M}_t} \setminus \{i\}} \widetilde{s}_{i,j} \log \frac{\widetilde{s}_{i,j}}{s_{i,j}}, \tag{3}$$

where $\lambda$ is a hyperparameter for controlling the emphasis on preserving representation relation.

To optimize the classifier weights for prediction on all tasks seen so far, we incorporate two additional terms: the classification loss and the distillation loss. For classification loss, we use the standard cross-entropy loss to distinguish concepts in both past and current tasks:

$$\mathcal{L}_{entropy} = \sum_{i \in \mathcal{B}_{\mathcal{D}_t} \cup \mathcal{B}_{\mathcal{M}_t}} y_i \log p_i, \tag{4}$$

where $p_i$ is the predicted probability of sample $i$ on the ground truth class $y_i$. The distillation loss is applied to the logits of memory samples to prevent information loss in the decision-level (Buzzega et al., 2020) and is given by:

$$\mathcal{L}_{distill} = \frac{1}{|\mathcal{B}_{\mathcal{M}_t}|} \sum_{i \in \mathcal{B}_{\mathcal{M}_t}} \|\boldsymbol{p}_i - \tilde{\boldsymbol{p}}_i\|_2^2, \tag{5}$$

where $\boldsymbol{p}_i$ and $\tilde{\boldsymbol{p}}_i$ are the logits of sample $i$ output by the fast and slow learner respectively.

The overall loss function then combines the classification loss and distillation loss, together with the proposed relaxed alignment loss and relation preservation loss:

$$\mathcal{L} = \mathcal{L}_{entropy} + \mathcal{L}_{distill} + \mathcal{L}_{relax} + \mathcal{L}_{preserve} \tag{6}$$

We optimize weights of the fast learner $\theta_F$ using the overall loss and update weights of the slow learner $\theta_S$ via an exponential moving average of $\theta_F$ after each optimization step at the rate of $\alpha \in [0, 1]$, *i.e.* $\theta_S \leftarrow \alpha\theta_S + (1 - \alpha)\theta_F$. This approach leverages the past knowledge encoded in the slow learner to guide the fast learner, ensuring that the performance on previous tasks remain stable as new knowledge is integrated. The gradual accumulation of knowledge into the slow learner also reduces abrupt changes in model weights that worsens forgetting. During inference, the slow learner is utilized to make predictions, maintaining consistency and reliability in the outputs.

## 4 PERFORMANCE STUDY

We evaluate our framework in two multi-modal continual learning scenarios, namely class incremental learning and domain incremental learning. In class incremental learning, new classes are introduced sequentially, requiring the model to adapt without forgetting previous knowledge. For domain incremental learning, the input data distribution shifts over time while the set of classes remains fixed, requiring the model to adapt to data from different domains. We measure the performance in terms of average accuracy across all steps, that is, $Accuracy_{all} = \frac{1}{T} \sum_{t=1}^{T} a_t$, where $a_t$ is the test accuracy on samples of all seen classes and domains after training at incremental step $t$. The results are averaged across three runs using different class or domain orders.

We use the following datasets for the experiments on class incremental learning:

- AVE (Tian et al., 2018): This is an audio-visual dataset consisting of 28 event classes including human activities, animal activities, music performance and vehicle sounds. We adapt this dataset for incremental learning by dividing these classes into seven different sequential steps, each comprising disjoint set of classes. Each class contains a minimum of 48 training videos and a maximum of 152 training videos, with each video spanning approximately 10 seconds.
- UESTC-MMEA (Xu et al., 2023): This is a egocentric dataset comprising of 32 activity classes, covering static activities such as watching television and reading, to physical activities such as walking and riding bike. For incremental learning, we randomly divide the classes into eight sequential steps. Each class contains a minimum of 129 training samples and a maximum of 171 training samples, with each sample having an average of 18 seconds of video recording and inertial data.

Table 1: Results of comparative study.

(a) Class incremental learning

| Method | AVE | UESTC-MMEA |
|--------|-----|------------|
| ER | $80.52_{\pm1.34}$ | $87.28_{\pm1.71}$ |
| A-GEM | $39.47_{\pm0.11}$ | $34.95_{\pm0.72}$ |
| DER++ | $82.40_{\pm1.71}$ | $90.45_{\pm1.45}$ |
| $Co^2L$ | $\underline{83.02}_{\pm2.15}$ | $89.74_{\pm1.28}$ |
| SSIL | $74.06_{\pm2.46}$ | $79.64_{\pm1.67}$ |
| AFC | $82.29_{\pm2.40}$ | $82.18_{\pm2.48}$ |
| ESMER | $80.37_{\pm1.81}$ | $\underline{91.01}_{\pm2.25}$ |
| AV-CIL | $66.55_{\pm0.24}$ | - |
| Ours | $\mathbf{89.66}_{\pm0.55}$ | $\mathbf{95.20}_{\pm1.03}$ |

(b) Domain incremental learning

| Method | KITCHEN | DKD |
|--------|---------|-----|
| ER | $64.54_{\pm1.13}$ | $72.69_{\pm0.41}$ |
| A-GEM | $65.38_{\pm1.14}$ | $70.31_{\pm0.87}$ |
| DER++ | $65.02_{\pm1.72}$ | $72.61_{\pm3.36}$ |
| $Co^2L$ | $65.30_{\pm1.40}$ | $73.26_{\pm1.17}$ |
| ESMER | $\underline{71.56}_{\pm3.44}$ | $\underline{75.56}_{\pm1.36}$ |
| UDIL | $62.40_{\pm3.47}$ | $67.04_{\pm1.60}$ |
| Ours | $\mathbf{74.81}_{\pm2.82}$ | $\mathbf{76.98}_{\pm0.62}$ |

For domain incremental learning, we create the following two datasets:

- KITCHEN: This dataset is derived from EPIC-KITCHENS-100 (Damen et al., 2022), a benchmark containing collection of audio-visual recordings of activities in different kitchen environments. We focus on ten commonly observed action classes, 'take', 'put', 'open', 'close', 'wash', 'cut', 'stir', 'pour', 'throw', 'move', and select the five environments with the largest number of training instances, namely 'P01', 'P02', 'P04', 'P22', 'P30'. For incremental learning, we introduce a different environment as the new domain at each step.

- DKD: This dataset is based on the diabetic kidney disease study in Betzler et al. (2023). It comprises retina images and tabular data containing patient information for the detection of diabetic kidney disease in three cohorts, namely SiDRP (Nguyen et al., 2016), SEED (Majithia et al., 2021) and SMART2D (Low et al., 2023). The variation in the disease prediction model's performance across the three cohorts suggests a significant distribution shift. We construct our sequential training dataset such that it has three incremental steps and each step introduces data from one of the three cohorts as the new domain.

We implement our framework in PyTorch (Paszke et al., 2019) and run all experiments on a single NVIDIA RTX A6000 GPU. We use a different encoder for each modality to ensure optimal feature extraction. For video data, we use the SlowFast network (Feichtenhofer et al., 2019), pretrained on Kinetics-400 (Kay et al., 2017) and designed to capture both spatial and temporal dynamics effectively. For image data in DKD, we use RETFound (Zhou et al., 2023), which is a foundation model trained on large-scale retinal images. For audio data, we use ResNet18 (He et al., 2016) that has been pretrained on the VGGSound dataset (Chen et al., 2020), allowing for robust audio feature extraction. For inertial data, we employ SSL-Wearables (Yuan et al., 2024), a self-supervised learning model trained on large-scale unlabeled wearable data to handle activity recognition tasks. We use two fully-connected layers, with 2048 hidden units, to fuse the representations from the modality-specific encoders to a 1024-dimensional joint representation before forwarding to a linear layer for the final classification output. More training details can be found in Appendix A2.

### 4.1 COMPARATIVE EXPERIMENTS

We compare our solution with the multi-modal continual learning method AV-CIL (Pian et al., 2023) and a range of continual learning methods, including ER (Riemer et al., 2019), A-GEM (Chaudhry et al., 2019), DER++ (Buzzega et al., 2020), $Co^2L$ (Cha et al., 2021), SS-IL (Ahn et al., 2021), AFC (Kang et al., 2022), ESMER (Sarfraz et al., 2023) and UDIL (Shi & Wang, 2024), that were originally designed and evaluated on single image modality datasets. We adapt these methods to handle multi-modality inputs by using the same pretrained encoders and fusion layers as our framework. For the memory size, we set $m = 100$ in AVE, UESTC-MMEA and KITCHEN, and $m = 10$ in DKD. We randomly select a balanced number of samples for each class and domain in all methods, except for AFC and ESMER which use specific selection strategy as indicated in their work.

Table 1(a) shows the results in the class incremental learning. Our solution demonstrates a significant improvement of 6.64% over the next-best-performing baseline $Co^2L$ in the AVE dataset. For

Table 2: Effect of removing individual loss components on AVE dataset.

| $\mathcal{L}_{relax}$ | $\mathcal{L}_{preserve}$ | $Accuracy_{all}$ | $Accuracy_{past}$ |
|:---:|:---:|:---:|:---:|
| ✓ | ✓ | $\mathbf{89.66}_{\pm 0.55}$ | $\mathbf{87.37}_{\pm 0.50}$ |
| ✗ | ✓ | $87.80_{\pm 0.94}$ ($\downarrow 1.86$) | $85.55_{\pm 2.06}$ ($\downarrow 1.82$) |
| ✓ | ✗ | $88.19_{\pm 0.88}$ ($\downarrow 1.47$) | $84.88_{\pm 1.27}$ ($\downarrow 2.49$) |
| ✗ | ✗ | $87.68_{\pm 0.53}$ ($\downarrow 1.98$) | $83.97_{\pm 0.43}$ ($\downarrow 3.40$) |

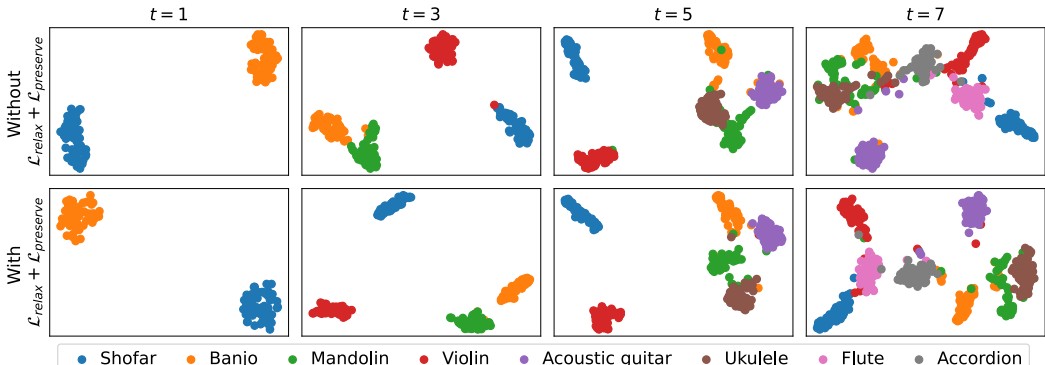

Figure 3: t-SNE visualization of representations from samples of musical instrument classes learned up to incremental step $t$ in AVE dataset, with and without both $\mathcal{L}_{relax}$ and $\mathcal{L}_{preserve}$.

the UESTC-MMEA dataset, we achieve a 4.19% higher accuracy than ESMER. This demonstrates the effectiveness of our model in consolidating and retaining knowledge from different modalities as new classes are introduced. Table 1(b) shows the performance of the various methods in the domain incremental learning. We achieve an improvement of 3.25% and 1.42% over the next-best-performing baseline ESMER in the KITCHEN and DKD datasets respectively. These results highlight our model's ability to effectively adapt to new domains while maintaining robust performance across tasks.

## 4.2 MODEL ANALYSIS

We conduct further analysis to gain insights on the effectiveness of our solution. We focus our analysis using the AVE dataset and comparison with baselines ER, DER++, Co$^2$L and ESMER.

**Ablation study on loss components.** We evaluate the effect of individual loss components on our model performance by systematically removing them from training. In addition to the average accuracy on all learned classes $Accuracy_{all}$, we also compute the average accuracy on samples of previously learned classes, that is, $Accuracy_{past} = \frac{1}{T-1} \sum_{t=2}^{T} u_t$, where $u_t$ is the test accuracy on samples of classes in $\bigcup_{i=1}^{t-1} \mathcal{Y}_i$ after training at incremental step $t$.

Table 2 summarizes the results. Our analysis shows that removing removing $\mathcal{L}_{relax}$ leads to a decrease of 1.86% and 1.82% in the average accuracy on all classes and old classes respectively, while $\mathcal{L}_{preserve}$ leads to a decrease of 1.47% and 2.49% respectively. The larger decrease in $Accuracy_{all}$ after the removal of $\mathcal{L}_{relax}$ highlights the critical role of multi-modal representation learning in adapting and integrating new knowledge. On the other hand, removing $\mathcal{L}_{preserve}$ results in a more significant drop in $Accuracy_{past}$, demonstrating its importance in maintaining the performance of previously learned classes. Removing both $\mathcal{L}_{relax}$ and $\mathcal{L}_{preserve}$ leads to largest performance drop of 1.98% and 3.40% in $Accuracy_{all}$ and $Accuracy_{past}$ respectively, indicating the importance of both losses in learning new knowledge and preserving old information.

To illustrate the effect of $\mathcal{L}_{relax}$ and $\mathcal{L}_{preserve}$ on representation learning, we visualize the sample representations when the model is trained with and without the two losses in Figure 3. We focus on

Table 3: Performance of relax loss $\mathcal{L}_{relax}$ vs. traditional contrastive loss $\mathcal{L}_{contrast}$ on AVE dataset.

|  | $Accuracy_{all}$ | $Accuracy_{past}$ |
|---|---|---|
| $\mathcal{L}_{relax} + \mathcal{L}_{preserve}$ | $89.66_{\pm 0.55}$ | $87.37_{\pm 0.50}$ |
| $\mathcal{L}_{contrast} + \mathcal{L}_{preserve}$ | $88.49_{\pm 0.13}$ ($\downarrow 1.17$) | $85.77_{\pm 0.32}$ ($\downarrow 1.60$) |
| $\mathcal{L}_{relax}$ only | $88.19_{\pm 0.88}$ | $84.88_{\pm 1.27}$ |
| $\mathcal{L}_{contrast}$ only | $86.43_{\pm 1.14}$ ($\downarrow 1.86$) | $81.97_{\pm 1.76}$ ($\downarrow 2.91$) |

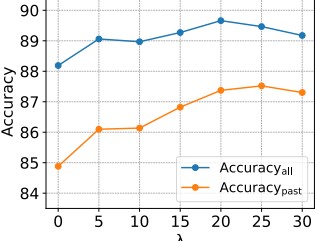
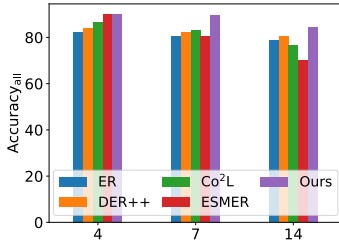
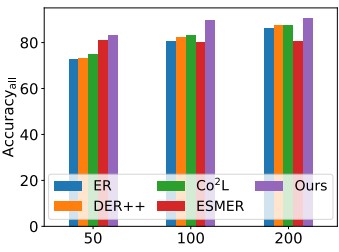

Figure 4: Effect of hyperparameter $\lambda$ on AVE dataset.

Figure 5: Results of on AVE dataset with different number of incremental steps $T$ and memory size $m$.

classes of similar domain, particularly musical instruments, that are learned across different incremental steps. We see that the representations trained with both losses are better clustered as more classes are learned, indicating their effectiveness in learning more robust representations. Particularly, when both losses are not used, there is increased confusion at the last incremental step $t = 7$ among the string instrument classes, namely Banjo, Mandolin and Ukulele. Such qualitative results shows that $\mathcal{L}_{relax}$ and $\mathcal{L}_{preserve}$ contributes to better separability among representations of similar object classes, thereby improving the model's classification performance.

**Effect of relaxing alignment constraint.** Table 3 presents the results when the traditional multi-modal supervised contrastive loss $\mathcal{L}_{contrast}$ (see Eq. A1 in Appendix A1) is used in place of our proposed $\mathcal{L}_{relax}$. We assess the effectiveness of $\mathcal{L}_{relax}$ under two conditions: with and without $\mathcal{L}_{preserve}$. The results show that using $\mathcal{L}_{contrast}$ leads to a performance decline in both cases. This suggests that by relaxing the constraint on cross-modality alignment, we enhance the robustness of multi-modal representations for incremental learning.

**Effect of hyperparameter $\lambda$.** We study how the hyperparameter $\lambda$ in Eq. 3, which controls the strength of $\mathcal{L}_{preserve}$, affects model performance. Figure 4 shows the average accuracy across all learned classes and previously learned classes for different values of $\lambda$. We see that performance is the worst when $\lambda = 0$, indicating the importance of $\mathcal{L}_{preserve}$. As $\lambda$ increases, there is a general improvement in performance, especially on the previously learned classes, which in turn improves the overall accuracy. However, the performance plateau when $\lambda$ exceeds 20. Although a higher $\lambda$ better preserves old knowledge, the overall accuracy would be negatively impacted as it hinders the learning of new knowledge.

**Effect of incremental steps $T$ and memory size $m$.** We examine the effect of number of incremental steps $T$ and memory size $m$ on the model performance compared to baseline methods. Varying total incremental steps $T$ directly affects the number of new classes introduced at each step as the classes in the dataset are divided equally into $T$ disjoint sets. Figure 5 shows that our solution demonstrates strong performance in long runs of small-sized tasks. We also see that our solution consistently outperforms the baselines under different memory size restriction.

**Recency bias.** One common issue in class incremental learning is recency bias, where model predictions are biased towards newly learned classes due to data imbalance. Figure 6 shows the prediction accuracy of the model at the end of training of each incremental step $t$ on samples of the

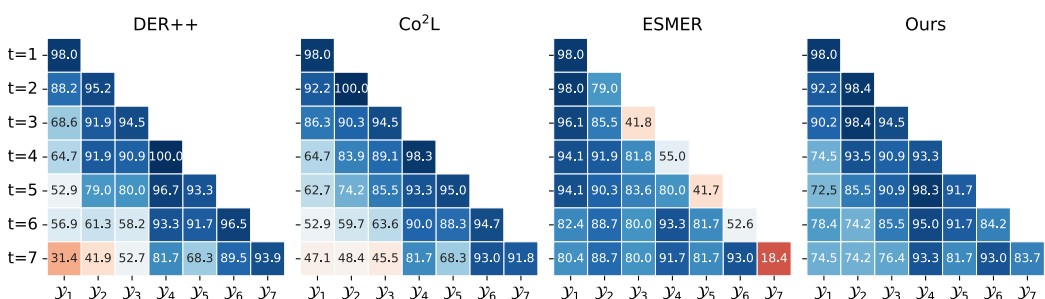

Figure 6: Accuracy on the seen classes ($x$-axis) after training at each step ($y$-axis) in AVE dataset.

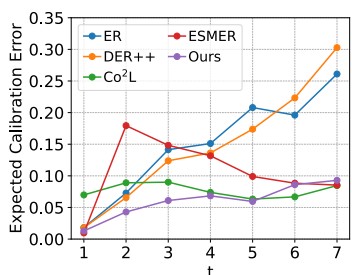

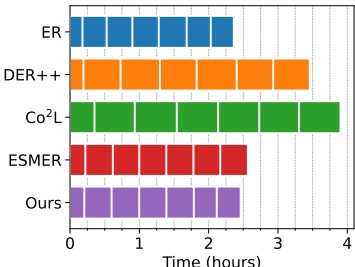

Figure 7: Expected calibration error after training at each incremental step $t$ on AVE dataset.

Figure 8: Average training time to complete all incremental steps on AVE dataset.

respective set of learned classes $\mathcal{Y}_1, \ldots, \mathcal{Y}_t$. The recency bias is observed in methods such as DER++ and Co²L. On the other hand, ESMER places a strong emphasis on maintaining the performance on old classes, which slows the learning of new classes. In contrast, our proposed approach achieves a better balance between learning of new classes and not forgetting old knowledge.

**Model calibration.** Model calibration measures how well a model's prediction confidence aligns with its actual accuracy. In other words, high confidence should indicate that the prediction is reliable while low confidence should suggest uncertainty. Poorly calibrated models are often an issue in continual learning as they tend to yield overconfident predictions on newly learned classes due to recency bias. To evaluate how well each model is calibrated, we compute the expected calibration error (ECE) (Guo et al., 2017), which is a weighted average of the difference between accuracy and confidence. Figure 7 shows the ECE value of each learner after training at each incremental step. We see that our solution has the overall lowest ECE value and remains relatively stable.

**Training time.** Another important consideration in continual learning is the overall training time required to assimilate new knowledge. Figure 8 shows the average training time incurred by the various learners to complete all the incremental steps on the AVE dataset. We see that our solution is efficient as its training time is comparable with respect to the simple replay method ER.

## 5 CONCLUSION

In this paper, we introduce a dual-learner framework for multi-modal continual learning, where multi-modal representation learning plays a crucial role. Our findings reveal that applying multi-modal supervised contrastive loss in continual learning leads to a decline in performance. To address this, we have proposed a new loss function to reduce information loss by relaxing the constraint on cross-modality representation alignment. We further mitigate forgetting by preserving consistency of the relation between previously learned representations. Extensive experiments across various continual learning scenarios and datasets involving different modalities demonstrate the effectiveness of our proposed solution in learning robust multi-modal representations, achieving good performance in acquiring new knowledge and retaining previously learned information.

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

APPENDICES

A1 PRELIMINARY EXPERIMENT USING TRADITIONAL CONTRASTIVE LOSS

Given a batch of $N$ training samples $\{(\{\boldsymbol{x}_i^k\}_{k=1}^K, y_i)\}_{i=1}^N$, we have the corresponding projected representations $\boldsymbol{z}_i^k$ for each input modality $\boldsymbol{x}_i^k$. Let $\mathcal{A}_i^k = \{\boldsymbol{z}_j^l \mid j \in [1,N], l \in [1,K]\} \setminus \{\boldsymbol{z}_i^k\}$ and $\mathcal{Q}_i^k = \{\boldsymbol{z}_j^l \in \mathcal{A}_i^k \mid y_i = y_j\}$ where $\mathcal{Q}_i^k$ contains representations of all modalities of samples in the batch with label $y_i$, excluding modality $k$ of sample $i$. Then the multi-modal supervised contrastive loss can be computed as:

$$\mathcal{L}_{contrast} = \frac{1}{NK} \sum_{i=1}^N \sum_{k=1}^K \frac{-1}{|\mathcal{Q}_i^k|} \sum_{\boldsymbol{q} \in \mathcal{Q}_i^k} \log \frac{\exp(\boldsymbol{z}_i^k \cdot \boldsymbol{q}/\tau)}{\sum_{\boldsymbol{a} \in \mathcal{A}_i^k} \exp(\boldsymbol{z}_i^k \cdot \boldsymbol{a}/\tau)}, \tag{A1}$$

where $\tau > 0$ is a scalar temperature hyperparameter and $\cdot$ denotes the cosine similarity between two normalized representations. Here, for each modality-specific representation $\boldsymbol{z}_i^k$ acting as an anchor, the loss aims to pull together all modality-specific representations $\boldsymbol{z}_j^l$ of samples with the same class label $y_i$ regardless of its modality $l$. Similarly, cross-modality representations of samples from different class are also included in the set of negative pairs.

We conduct a preliminary experiment to examine the effectiveness of the contrastive loss on multi-modal continual learning. We use the AVE dataset with total incremental steps $T = 7$ and memory size $m = 100$. Using the same model architecture as described in Section 4, we train two variants of the model: one optimized using only cross-entropy loss, and the other using combination of cross-entropy loss and the multi-modal supervised contrastive loss in Eq. A1.

Table A1 summarizes the average accuracy $Accuracy_{all}$ achieved by the two models on the learned classes across all incremental steps, while Figure A1 shows the accuracy achieved at each incremental step $t$ on all classes learned so far and only on classes learned in previous steps. We see that performance of the model trained with the contrastive loss degrades faster as learning progresses, especially on the old classes. This suggests that the contrastive loss accelerates the forgetting of previously learned knowledge.

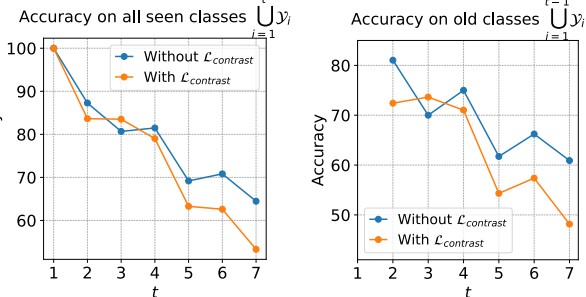

| | $Accuracy_{all}$ |
|---|---|
| Without $\mathcal{L}_{contrast}$ | 79.13 |
| With $\mathcal{L}_{contrast}$ | 75.05 |

Table A1: Average accuracy over all steps using model trained with and without $L_{contrast}$ on AVE dataset.

Figure A1: Accuracy at each time step on all seen classes and only on old classes using model trained with and without $L_{contrast}$ on AVE dataset.

## A2 TRAINING DETAILS

Table A2 shows the hyperparameters used for the respective baseline methods and our method in each experiment. For all these methods, we only fine-tune the last $F$ blocks of the pretrained encoders throughout the incremental learning steps. Particularly, we set $F = 1$ for the SlowFast and VGGSound encoders, and $F = 2$ for the SSL-Wearables and RETFound encoders. We optimize the models for 20 epochs using Adam optimizer at a learning rate of $1e^{-4}$. As for the multi-modal continual learning baseline AV-CIL, we use their proposed model architecture and the hyperparameter values they provided for AVE.

Table A2: Hyperparameters used in our experiments.

| Dataset | Method | Hyperparameters |
|---|---|---|
| AVE | ER | $bs = 16$ |
| | A-GEM | $bs = 16$ |
| | DER++ | $bs = 16, \alpha = 0.5, \beta = 0.5$ |
| | Co$^2$L | $bs = 32, \tau = 0.1, \kappa = 0.2, \kappa^* = 0.01, \lambda = 1.0$ |
| | SSIL | $bs = 16, \tau = 2$ |
| | AFC | $bs = 16, \lambda_{disc} = 1.0$ |
| | ESMER | $bs = 16, \alpha_l = 0.99, \beta = 1.0, \alpha = 0.999, \gamma = 0.15, r = 1.0$ |
| | Ours | $bs = 16, \lambda = 20, \eta = 0.07, \alpha = 0.997$ |
| UESTC-MMEA | ER | $bs = 16$ |
| | A-GEM | $bs = 16$ |
| | DER++ | $bs = 16, \alpha = 0.5, \beta = 0.5$ |
| | Co$^2$L | $bs = 32, \tau = 0.1, \kappa = 0.2, \kappa^* = 0.01, \lambda = 1.0$ |
| | SSIL | $bs = 16, \tau = 2$ |
| | AFC | $bs = 16, \lambda_{disc} = 4.0$ |
| | ESMER | $bs = 16, \alpha_l = 0.99, \beta = 1.0, \alpha = 0.999, \gamma = 0.15, r = 1.0$ |
| | Ours | $bs = 16, \lambda = 20, \eta = 0.07, \alpha = 0.998$ |
| KITCHEN | ER | $bs = 16$ |
| | A-GEM | $bs = 16$ |
| | DER++ | $bs = 16, \alpha = 0.5, \beta = 0.5$ |
| | Co$^2$L | $bs = 32, \tau = 0.1, \kappa = 0.1, \kappa^* = 0.1, \lambda = 1.0$ |
| | ESMER | $bs = 16, \alpha_l = 0.99, \beta = 1.0, \alpha = 0.999, \gamma = 0.15, r = 0.1$ |
| | UDIL | $bs = 16, \lambda_d = 0.5, C = 5, lr_{task} = 2e^{-3}, lr_{discriminator} = 1e^{-5}$ |
| | Ours | $bs = 16, \lambda = 5, \eta = 0.07, \alpha = 0.9999$ |
| DKD | ER | $bs = 16$ |
| | A-GEM | $bs = 16$ |
| | DER++ | $bs = 16, \alpha = 0.5, \beta = 0.5$ |
| | Co$^2$L | $bs = 32, \tau = 0.5, \kappa = 0.2, \kappa^* = 0.1, \lambda = 1.0$ |
| | ESMER | $bs = 16, \alpha_l = 0.99, \beta = 1.0, \alpha = 0.999, \gamma = 0.15, r = 1.0$ |
| | UDIL | $bs = 16, \lambda_d = 0.5, C = 5, lr_{task} = 2e^{-3}, lr_{discriminator} = 1e^{-5}$ |
| | Ours | $bs = 16, \lambda = 20, \eta = 0.07, \alpha = 0.9995$ |

