# OpenReview forum: "Relaxing Representation Alignment with Knowledge Preservation for Multi-Modal Continual Learning"
_ICLR.cc/2025/Conference — ICLR 2025 Conference Withdrawn Submission_

### Official Review · Reviewer_BSpD · 2024-10-31

**Soundness:** 2
**Presentation:** 2
**Contribution:** 2
**Rating:** 5
**Confidence:** 4

**Summary:**

This work presents a technique to work on the continual learning problem in the multi-modal setting. In particular, this work proposes a relaxed cross-modality representation alignment loss and employs a dual-learner framework to maintain the relationships between previously learned representations. Using this framework, the model can alleviate the catastrophic forgetting issue in multi-modal continual learning. The efficacy of the proposed method is evaluated on various continual learning tasks and benchmarks, namely AVE, UESTC-MMEA, KITCHEN, and DKD. It is proven that the proposed approach could improve overall performance on these benchmarks.

**Strengths:**

- The proposed method could achieve notable results across datasets. Compared to the SOTAs, the proposed method can beat the best performance of past techniques by up to 6%. This shows that the proposed method is proven effective in the multi-modal continual learning setting.
- This work also provides ablation studies to understand the proposed method and its components in greater detail. The visualization of feature embeddings is presented with the proposed losses. Also the work provides results with varying incremental steps and the memory size.

**Weaknesses:**

- This work lacks novelty by using the relaxed contrastive loss which in fact is quite similar to a common multimodal contrastive loss. Take an example of the well-known  Contrastive Language-Image Pre-Training (CLIP) model, the multimodal contrastive loss is already presented in this work, though the modalities are different. Also, the contrastive loss is not connected directly to the continual learning problem. The contrastive loss is more intended to align between representations to boost up overall performance rather than retaining past knowledge.
- The other components and losses have been adopted from past works in continual learning e.g., Li et al., “Learning without forgetting” and Hou et al. “Learning a Unified Classifier Incrementally via Rebalancing”. The weights update allows two separate entities of fast and slow learners. The novelty of this approach is also limited as the previous work has already presented this approach (see Simon et al., “On Generalizing Beyond Domains in Cross-Domain Continual Learning”).
- The main contribution of this work is multimodal continual learning. However, except the relaxed contrastive loss, the other components do not directly contribute to multimodal continual learning problems. For instance, the paper can discuss the impact of distillation in the mixed feature from two modalities compared to only one. In the current state, the focus of this paper is not quite clear rather than obtaining better results in continual learning.
- In experiments, we need to also observe forgetting factors caused by acquiring new information. However, there is no such discussion and performance analysis in the experimental section. Assessing the model's effectiveness only through accuracy might be misleading, as the observed performance gains could be attributed to the introduced contrastive loss, while the forgetting score (i.e., the performance drop between tasks) might remain unaffected.

**Questions:**

Please answer concerns in the weaknesses.

---

### Official Review · Reviewer_dLR5 · 2024-11-01

**Soundness:** 3
**Presentation:** 2
**Contribution:** 2
**Rating:** 5
**Confidence:** 3

**Summary:**

This paper addresses the issue of continual learning in the multi-modal context. For the 'continual learning' part, it employs a dual learner framework, where a fast learner captures the current knowledge and a slow learner accumulates (and maintains) learned experience.  For the "multi-modal" part, the authors propose a relaxed contrastive learning method that aims to mitigate the loss of modality-specific information in the standard contrastive learning. Experiments on various datasets demonstrate the method's effectiveness in both task-incremental and domain-incremental situations. This paper also includes a detailed ablation study on the components of the proposed method.

**Strengths:**

- This work focuses on the important topic of multi-modal continual learning. As far as I am concerned, the proposed relaxed contrastive learning scheme is novel and beneficial. Through integrating this scheme with a two-learner framework, the study effectively shows performance improvements in both task- and domain-incremental scenarios.
- The basic modeling ideas are clearly articulated and offer insights for future research.
- This paper includes a detailed ablation study on the components of the proposed method.

**Weaknesses:**

- The investigation into representation learning is inadequate.  The clustering analysis provide little information about how the proposed method improve the learned representations.
    - as this work is not built upon large-scale experiments (say on 10 or 15 datasets), I would expect to see more concrete and in-depth analysis from the representation learning aspect to support the key ideas of their modeling.
- The ablation study on loss functions only addresses $L_{\text{relax}}$ and $L_{\text{preserve}}$, omitting other potentially significant aspects such as $L_{\text{distill}}$, which could be crucial for aligning currently learned representations with previous experience.
- The current method only constructs the memory dataset $\mathcal{M}_t$ naively, while the quality of $\mathcal{M}_t$ should be essential for the proposed method, potentially limiting its application to more general scenarios.
- The presentation needs improvement; for instance, Fig. 3 does not clarify which representation is visualized, and the details of the network parameters are ambiguous: are there two hidden layers with 2048 neurons each or 2048 neurons in total?
- The evaluation of model performance may be biased in favor of the proposed method, as evidenced by the ESMER method, which appears less impressive primarily due to its low accuracy on the latest task. For further details, refer to the Questions section.

**Questions:**

One main concern is that does the model indeed captures the modality-specific information? and how much in the performance improvement can be attribute to this part?
- despite applying the relaxed contrastive learning scheme, there is no direct driving force to ensure the model learns modality-specific information. That is modality-specific components may not be reflected in the joint representation.
- Does Fig.3 visualize the representation $\tilde{z}$?
- In Fig.3, the difference in clustering patterns in two situations is subtle. In addition, it is more relevant to compare representations trained with $L_{\text{contrast}}$ and $L_{\text{relax}}$ . And the visualization need not to be restricted to $\tilde{z}_i$, the behavior of other intermediate representations such as $z^k_i$ also provide valuable information.
- In Fig.3, it would make more sense to project representations at different step $t$ to a common subspace instead of finding a new subspace at each step.
- What is difference in similarity score (like the one in Fig.1) when using  $L_{\text{relax}}$ instead of  $L_{\text{contrast}}$?


The other significant concern is whether the model retains representations of previously learned tasks.
- Aside for the accuracy, there is no direct evidence indicating the quality of the preservation of representations on learned tasks. The temporal behavior of $L_{\text{preserve}}$ and $L_{\text{distill}}$ could be informative. One can compare their behaviors when trained with $L_{\text{relax}}$ and $L_{\text{contrast}}$. One may also use similarity score to evaluate the preservation of representations on learned tasks.
- In my view, both $L_{\text{preserve}}$ and $L_{\text{distill}}$ shape the available area in the feature space for continual learning. Ablation on $L_{\text{distill}}$ should be include. The interplay of this two loss functions with $L_{\text{relax}}$ is also of interest.
- The proposed continual learning method is rehearsal based. Therefore the memory dataset $\mathcal{M}_t$ should be essential for its performance in maintaining knowledge of learned tasks/domains. Currently,  $\mathcal{M}_t$ is constructed naively (randomly selected with balanced number of samples for each task/domain) and the ablation study only concerns about the memory size. When the method is applied to more general scenarios (say additional datasets, more complex tasks, etc.) the quality of $\mathcal{M}_t$ may become a bottleneck. Can you outline your solution to this potential challenge?
- How much is the performance depends on $\alpha$? Do you try other ways to update the slow learner?
- Does the order of tasks/domains matter during the continual learning process?
- Can you discuss about the upper bound of the number of tasks/domains in your method?
- It is interesting to ask that how or how much do the representation on learned tasks being reused during the learning of new ones.


Other questions:
- The performance of ESMER (Fig.6) is poor only on the current task, but this accuracy will be considerably improved in the next step, while maintaining good performance on previous tasks. I wonder if one can add a dummy step after current task, i.e., repeating the current task once more so that their model will have decent performance on all tasks?
    - A related concern is that the metric in table 1 (i.e., $Accuracy_{all}$) could be biased. Evaluating using $Accuracy_{past}$ may yield a higher rank for ESMER.
- The main text frequently mentions "robust representations" (e.g., lines  95, 103, 137, etc.). What is meant specifically by "robust"?
- In line 121, you mention "i.i.d. multi-modal classification tasks". In what sense are you using the term 'i.i.d.'?

---

### Official Review · Reviewer_K15L · 2024-11-03

**Soundness:** 2
**Presentation:** 3
**Contribution:** 2
**Rating:** 3
**Confidence:** 4

**Summary:**

This paper employs a dual-learner framework for multi-modal continual learning. On one hand, the authors introduce a relaxed representation alignment loss to encourage the model to retain diverse features captured by different modalities. On the other hand, they constrain the outputs of the fast learner and slow learner to mitigate catastrophic forgetting. The proposed method demonstrates considerable improvements over the state-of-the-art results across various benchmarks.

**Strengths:**

1.	This paper is well-organized and clearly written, making it easy to understand.
2.	The proposed method is simple yet effective, as shown by both quantitative experiments and qualitative analysis that validate its performance.

**Weaknesses:**

1.	This paper lacks a clear research motivation, and some claims in the paper appear contradictory. For instance, the abstract suggests that multi-modal continual learning is more challenging than single-modal continual learning, which is more commonly explored. Conversely, in the introduction, the authors claim that multi-modal data offer significant advantages for continual learning. Critically, both the challenges and benefits of multi-modal continual learning are not sufficiently clarified, raising questions about the necessity and significance of investigating multi-modal continual learning.
2.	In the experimental section, the authors only compare their method with AV-CIL on the AVE dataset under class-incremental setting. While AV-CIL is specifically designed for multi-modal continual learning, the other methods compared are solely for single-modal continual learning. This is especially notable given that the authors have already mentioned several recent proposed methods in the related work.
3.	Although the authors provide a detailed description of the datasets, it is unclear how these datasets were divided into class-incremental tasks for most experiments, except for the ablation study shown in Fig. 5.
4.	The ablation study on loss weights is insufficient. The authors only conducted ablation experiments on the preserve loss and relax loss, and provided an explanation for determining the weight of the former. However, there is no ablation study for the distill loss, and it is entirely unclear how the authors set the weights for the relax loss and distill loss.
5.	There has been extensive prior work attempting to develop dual-learner frameworks in continual learning. Could the authors explain how this work differs from previous efforts and specify the improvements and innovations it introduces?

**Questions:**

1.	On line 105, in the sentence “where data of different distributions or classes become available over time,” it seems that “available” should be “unavailable.” Otherwise, this statement would contradict the definition of incremental learning.
2.	Since the main text has a maximum length of 10 pages, it would be beneficial for the authors to move the experiments from the supplementary materials to the main text, making the paper more cohesive and complete.

---

### Official Review · Reviewer_Ltfe · 2024-11-04

**Soundness:** 2
**Presentation:** 3
**Contribution:** 2
**Rating:** 5
**Confidence:** 3

**Summary:**

This paper proposes a relaxed cross-modality representation alignment loss and utilize a dual-learner framework to preserve the relatoin between previously leraned representations. Experiments are conducted on several multi-modal datasets that encompass various types of input modalities. Results show that the proposed method consistently outperform baseline continual learning methods in both class and domain incremental learning scenarios.

**Strengths:**

The experimental results are significant, demonstrating that this method outperforms existing approaches in various continual learning scenarios, such as class-incremental learning and domain-incremental learning.

**Weaknesses:**

1. In continual learning, forgetting is a crucial evaluation metric that should be compared. A commonly used metric for evaluating forgetting is backward transfer, as mentioned in [1]. It is recommended to compare multiple metrics in the comparative experiments, i.e., in Table 1.
2. The dual framework and contrastive learning have been applied in many works like MoCo[2], Co2l[3], and AV-CIL[4]. It appears that this paper primarily extends previous work incrementally and may not present a sufficiently novel contribution.

[1] Lopez-Paz D, Ranzato M A. Gradient episodic memory for continual learning[J]. Advances in neural information processing systems, 2017, 30.

[2] He K, Fan H, Wu Y, et al. Momentum contrast for unsupervised visual representation learning[C]//Proceedings of the IEEE/CVF conference on computer vision and pattern recognition. 2020: 9729-9738.

[3] Cha H, Lee J, Shin J. Co2l: Contrastive continual learning[C]//Proceedings of the IEEE/CVF International conference on computer vision. 2021: 9516-9525.

[4] Pian W, Mo S, Guo Y, et al. Audio-visual class-incremental learning[C]//Proceedings of the IEEE/CVF International Conference on Computer Vision. 2023: 7799-7811.

**Questions:**

Could the authors explain in detail how the encoder outputs are fused in the fusion layers and the specific structure of the fusion layers?

---

### Note · Authors · 2024-11-14

I have read and agree with the venue's withdrawal policy on behalf of myself and my co-authors.